# RANBP1 (RAN Binding Protein 1): The Missing Genetic Piece in Cancer Pathophysiology and Other Complex Diseases

**DOI:** 10.3390/cancers15020486

**Published:** 2023-01-12

**Authors:** Salvatore Audia, Carolina Brescia, Vincenzo Dattilo, Lucia D’Antona, Pierluigi Calvano, Rodolfo Iuliano, Francesco Trapasso, Nicola Perrotti, Rosario Amato

**Affiliations:** 1Dipartimento di Scienze della Salute, Campus Salvatore Venuta, Università degli Studi “Magna Graecia” di Catanzaro, Viale Europa, 88100 Catanzaro, Italy; 2Dipartimento di Medicina Sperimentale e Clinica, Campus Salvatore Venuta, Università degli Studi “Magna Graecia” di Catanzaro, Viale Europa, 88100 Catanzaro, Italy

**Keywords:** RANBP1, SGK1, tumor, nuclear transport, mitotic stability

## Abstract

**Simple Summary:**

In just less than 30 years since the cloning of RANBP1 (RAN binding protein 1), this review represents the first attempt for a comprehensive and analytical compilation of the main works that have characterized the history of this gene. Originally defined as an accessory element of not well-defined proteins involved in nuclear transport, RANBP1 has been demonstrated, over time, to play an essential role in maintaining both nuclear import/export machinery and spindle check-point formation. Since the first biochemical characterization, knowledge has accumulated that paints a very complex picture of how RANBP1 and its Small-GTPase-related complex plays a fine regulation of cellular homeostasis, with a differentially phase-cell-dependent role. From its molecular context, up to the role in neoplastic and other pathologies of complex traits, RANBP1 always embodies further the missing piece of a gene signature that recapitulates several aspects of mammalian cell biology.

**Abstract:**

RANBP1 encoded by *RANBP1* or *HTF9A* (Hpall Tiny Fragments Locus 9A), plays regulatory functions of the RAN-network, belonging to the RAS superfamily of small GTPases. Through this function, RANBP1 regulates the RANGAP1 activity and, thus, the fluctuations between GTP-RAN and GDP-RAN. In the light of this, RANBP1 take actions in maintaining the nucleus–cytoplasmic gradient, thus making nuclear import–export functional. RANBP1 has been implicated in the inter-nuclear transport of proteins, nucleic acids and microRNAs, fully contributing to cellular epigenomic signature. Recently, a RANBP1 diriment role in spindle checkpoint formation and nucleation has emerged, thus constituting an essential element in the control of mitotic stability. Over time, RANBP1 has been demonstrated to be variously involved in human cancers both for the role in controlling nuclear transport and RAN activity and for its ability to determine the efficiency of the mitotic process. RANBP1 also appears to be implicated in chemo-hormone and radio-resistance. A key role of this small-GTPases related protein has also been demonstrated in alterations of axonal flow and neuronal plasticity, as well as in viral and bacterial metabolism and in embryological maturation. In conclusion, RANBP1 appears not only to be an interesting factor in several pathological conditions but also a putative target of clinical interest.

## 1. Introduction

RANBP1 is encoded by a gene named either *RANBP1* (ENSG00000099901) or *HTF9A* (Hpall Tiny Fragments Locus 9A), allocated on chromosome 22 at the 22q11.21 locus with the approximate size of 11.418 bases [1]. Sixty-three different regulatory elements for RANBP1, such as promoters and enhancers, have been identified along its sequence, among which GH22J020114 and GH22J020113 are more frequently associated with its sequence [2]. The protein structure contains a similar to the pleckstrin homology (PH) domain and defined BD1-domain, as it is involved in protein–protein interaction with RAN and in cell signaling [3]. *RANBP1* counts multiple pseudogenes encoded on chromosomes 17, 9, 12 and X, showing a wide variety of alternative splicing, resulting in multiple transcript variants [4]. All these genetic and structural information are depicted in Figure 1.

The crucial molecular functions ascribed to this gene, as a member of the RAN-network and ascribed to the RAS superfamily of small GTPases, are: (i) taking action in RAN-dependent nucleo-cytoplasmic transport [5]; (ii) interfering in TNPO1-dependent inhibition of RAN GTPase activity and mediating the dissociation of RAN from proteins involved in transport within the nucleus [3]; (iii) to induce a conformational change in the XPO1-RAN complex, thus leading to the release of the nuclear export signal [6]; (iv) favoring the disassembly of the complex formed by RAN and importin beta [7]; (v) promoting the dissociation of RAN from the complex instated with KPNA2 and CSE1L [3]. More recently, a role of RANBP1 has been assessed in the regulation of mitotic spindle assembly and progression of mitosis [8]. Essentially, RANBP1 does not increase RAN GTPase activity per se, but it plays as a trans-activator of the RANGAP1-mediated GTP hydrolysis and inhibits an RCC1-dependent exchange of GDP-bound RAN with GTP [9,10].

RANBP1 displays a low tissue specificity (RNA) but at the same time exhibits a high distribution of tissue expression. RANBP1 is an intracellular protein with a strong cytoplasmic expression, especially in the testis, intestinal tract, lymphoid and hematopoietic tissues and squamous epithelia [11,12]. The transcript has been detected in human, mouse, and pig brains showing, generally, a low regional specificity [13]. Interestingly, a peak in the human cerebral cortex, such as the ventromedial prefrontal cortex, has been described [14]. RANBP1 is present in all immune cell lines, with a high specificity in T and B-cells [15]. The Immunoassay or Proximity Extension Assay (PEA) failed to detect RANBP1 in blood, whereas it can constantly be isolated by mass spectrometry methods [16].

## 2. Role of RANBP1 in Nuclear Transport

Unlike what was learned from the lecture on fungal biology, in animal cells RANBP1-dependent regulatory mechanisms show significant differences. The mechanism of CRM1-RanGTP cargo sequestration differs considerably between fungi and animal cells; in fact, in the latter, the RANBP1 is exclusively involved in RANGTP sequestration from the nuclear export complex. In fungi, RANBP1, CRM1 and RANGTP form a 1:1:1 nuclear export complex; in contrast, animal RANBP1, CRM1 and RANGTP set a 1:1:2 ratio. This is almost attributable to a different affinity of complex RANBP1-RANGTP-CRM1, probably mediated by the interspecies of the non-conservation of the amino acid residues responsible for their interaction [17].

As a rule of thumb, the GTP-related proteins occur bound to GTP or GDP. GTPases are provided with hydrolytic activity toward guanoside nucleotide. The exchange between the two forms is enabled, in vivo, by the increase in RANGAP1-mediated hydrolytic power, in the time interval when RCC1 recharges GTP on RAN. Simultaneously, an additional noncatalytic component (RANBP1) intervenes through its ability to bind only the GTP-bound form of RAN [9]. In brief, RANBP1 makes RANGTP accessible to RANGAP1-mediated hydrolysis. The effect of this interaction is to exponentially enhance the hydrolysis efficiency on RAN-GTP. In addition, RANBP1 inhibits the exchange activity of RCC1, at least by in vitro evidence [10]. RANBP1 is a RAN major effector for the ability to interact only with RANGTP, but not with RANGDP, thus inducing a dramatic switch in the RAN functional state. Moreover, RANBP1 plays a crucial role in RAN-dependent nucleo-cytoplasmic transport. A small nuclear G protein-1 has been identified [9] and subsequently cloned from the rat [10], proving be identical to a previously identified open reading frame (HTF9A) [18]. In first experimental evidence, the gene is demonstrated being a co-activator of RANGTPase, able to complex with RANGTP and RCC1-RAN, holding the latter complex in a less responsive state for guanine nucleotide exchange on RAN. RANBP1 thus inhibits the RCC1-dependent GTP exchange on RAN [10]. The experimental evidence supports the existence of separate domains in the RAN protein: one dependent on RANBP1, essential for cell cycle progression and RNA export, the other one independent from RANBP1, that influences the nuclear protein import [19]. Moreover, the RANBP1-RAN binding contributes to stabilize the RAN-P97 (importin capable of recognizing nuclear localization sequences-NLS) interaction, thus promoting the protein import through the nuclear pore complex [20,21]. RANBP1, which is bound to RANGTPase, intervenes in both steps of nuclear protein import: (i) docking to the nuclear pore complex and ATP-dependent translocation through the nuclear envelope; (ii) enhancing a link between the binding and translocation steps. RANBP1 acts upon the nuclear export signals (NES) specialized protein-domains, required for cytosolic localization of proteins. Indeed, following the rupture of the nuclear envelope (M phase initiation of mitosis), the soluble nuclear and cytoplasmic components collapse together. When the nuclear membrane reforms, RANBP1 is exported, abutting to the cytoplasmic side of the nascent nuclear membrane, whereas most of the cellular RAN accumulates into the nucleus. The RANBP1 release from the nucleus is conditioned by a NES, to prevent that due its small size, it would passively diffuse into the nucleus, thus causing RAN nuclear sequestering and the inhibition of protein import [22,23]. The abatement of RANBP1, in *Xenopus* egg extract samples, resulted in the decrease in RCC1 as well, while the deprivation from the samples of both components revealed no abnormalities. On the other side, altering the protein ratio, defects were observed in DNA replication, nuclear assembly and nuclear transport of proteins. It follows from this evidence that the pivotal point for normal cell cycle progression is the balance of the RANBP1/RCC1 ratio [24]. Concurrently, the role of RANBP1 in the mechanism of the RANGTP- transport receptor (e.g., importin beta) complex disassembly was confirmed, thus defining a role as a catalytic processor in GTPase fluctuations, resulting in GTP hydrolysis [7]. Moreover, ARAN1 (monoclonal antibody against human RAN, recognizing an epitope in the COOH-terminal domain), when injected into cultured cells, inhibited protein import into the nucleus by suppressing RANBP1 binding to the RAN-Importin beta complex [25]. In conclusion, this evidence made it possible to infer that the binding of RANBP1 to the RAN-Importin beta complex is essential and limiting for the complex dissociation into the cytoplasm required for RAN release and for its subsequent recycling and nuclear reuse. A study of particular interest, starting from the use of leptomycin B, as a nuclear export inhibitor, has led to a revision of the knowledge for physiological RAN-RANBP1 nuclear transport mechanism. The nuclear accumulation of RANBP1 was obtained, demonstrating that transport through nuclear pores does not occur by passive diffusion, as the small size of the molecule would have suggested. It is inferred that the translocation of RANBP1 occurs by an active mechanism that requires the involvement of RANGTP [5]. The shuttling of RANBP1 may be used to clean the nuclear pores of RANGTP, thus preventing the premature release of protein cargoes from nuclear transport complexes (Figure 2).

Experiments by practicing a RANGAP-RANBP1 nuclear injection, produce a blocking of the tRNAs export, but not that of spliced mRNAs, thus demonstrating that these might use a RAN-independent export pathway [27]. However, it should be mentioned that this could be a variable, species- and tissue-specific mechanism. Of particular note should be mentioned the RANBP1-CRM1 interaction, during cytoplasmic proteins’ and ribonucleoproteins’ release. An allosteric mechanism has been identified, whereby the trimeric complex formed by CRM1-RANBP1-RANGTP may assume a conformation such that CRM1, per se, is able to release the protein cargo for export, but unable to bind a new cargo in the absence of RANGTP [6]. Recently, it has been demonstrated that RANBP1 is essential and limiting in the transport of pre-microRNAs, by modulating the action of RAN and RANGAP1 and ensuring the maintenance of a nucleus-cytoplasmic GTP/GDP gradient, which allows the export of pre-miRNA, in assembly to Exportin-5, thus ensuring their cytoplasmic influx and maturation [28]. The maintenance of the GTP/GDP gradient across the nuclear membrane is influenced by SGK1 fluctuations through the Sp1-dependent transcriptional regulation of RANBP1 and RANGAP1. This mechanism has pleiotropic features and has been verified in both normal primary and tumor cell lines and appears to function as a general epigenomic mechanism (Figure 3).

## 3. Role of RANBP1 in Mitotic Stability

During prometaphase, when the nuclear membrane undergoes involution processes, the protein network that is involved in nuclear transport is not affected by degradation processes, as previously believed, but instead precipitates on nascent kinetochores in the early steps of their formation, contributing to the fate of cell mitosis [29].

Early evidence on the RANBP1 role in mitotic stability can be traced back to studies on the murine form of the gene. Mouse *Htf9* gene, encoding for Ranbp1, through its interaction with the Ras-related protein Ran, is implicated in several nuclear events, such as DNA replication, RNA export, protein import, and monitoring of the completion of DNA synthesis, prior to the initiation of mitosis. Several links were identified between *Htf9*/*RanBP1* gene expression and cell proliferation, pointing to the evidence that this gene exerts transcriptional control during the cell cycle [30]. *RanBP1* overexpression interferes with the orderly execution of the cell division at different stages and causes DNA replication inhibition [31]. The expression of the murine *Htf9-C*/RanBP1 gene is regulated by a bidirectional promoter and is downregulated in cells arrested in G0, whereas it is progressively increased in G1/S phase, reaching maximum activity in S phase [32]. *RANBP1* is transcribed in a cell cycle-dependent manner. Indeed, within the RanBP1 promoter sequence, two binding sites for E2F factors (cell cycle regulatory transcription factors) are present. These factors are: E2F-b, which is necessary for the positive transcriptional regulation during the S-phase and E2F-c, which is required for the transcriptional repression during cell growth arrest. DNA binding site for E2F-c is closer than E2F-b to the transcription starting site of the gene, and this allocation also reflects their different functional roles. The binding sites for E2F-b and E2F-c drive opposite genetic functions and regulate the *RanBP1* transcription, through distinct molecular mechanisms [33]. During mitosis in mammalian cells, GTP-bound RAN (RAN-GTP) is clustered near mitotic chromosomes, while GDP-bound RAN (RAN-GDP) is distal to chromosomes. This pattern spatially regulates the spindle assembly, because RAN-GTP locally enables the release of spindle assembly factors (SAFs), such as Hepatoma Up-Regulated Protein (HURP), from their inhibitory interactions [34]. At this point, the rise of RANBP1 perturbs the mitosis by playing as inhibitor, and disrupts the positive balance of the triphosphate guanioside nucleotides of RAN. RANBP1 dramatically alters M-phase spindle assembly [35]. Confirmation of the gene’s influence, during mitosis, comes from the evidence that RANBP1 regulates mitotic spindle assembly, controls microtubule dynamics along metaphase and anaphase and nuclear chromatin reorganization following the cell’s exit from mitosis. RANBP1, thus following a cyclical pattern, increases from S phase to M phase, reaches a peak in metaphase and then sharply decreases during telophase. Interestingly, *RANBP1* overexpression is considered responsible for abnormalities in mitosis and defects in spindle assembly [36]. High levels of Ran exchange factor (RCC1) suppress the spindle-assembly checkpoint, at least in *Xenopus* egg extracts, by disrupting regulatory proteins during kinetochore positioning. In contrast, the RanGAP1/RanBP1 addition to the extracts restores spindle checkpoints, thus demonstrating the dependence on RanGTP level fluctuations [37]. More fine-grained and biochemically determined is the demonstration that RANGAP1 phosphorylation, especially on Ser358, is responsible for the stabilization of RANGAP1/RAN/RANBP1 interaction. Site-direct mutagenesis, by nullifying Ser358 phosphorylation, reduces the stability of this interaction. Since RANGAP1 behaves as a target of both the kinetochore and the spindle, it becomes clear how the mechanism may affect mitosis [38]. The further deepening of the previous studies ascertains that RANBP1, by coordinating the proper functioning of a number of mitosis-regulating factors, within the spindle-microtubules, ensures proper human chromosome segregation and normal progression of mitosis. On the contrary, *RANBP1*-depleted cells undergo a delay during prometaphase transition, often followed by apoptosis [39]. While the expression of the other members of RAN’s network is constant during mitosis, the expression of RANBP1 is modulated during mitosis [40]. Posttranslational modifications of the protein also play an important role in mitosis control. In fact, PLK1-dependent RANBP1 phosphorylation is essential for its interaction with RAN and for microtubule nucleation and spindle assembly during mitosis [41]. The role of RANBP1 in mitotic progression reaches a definition point with the discovery that, in the RCC1/RAN/RANBP1 complex, the latter plays a starring role in controlling the RanGTP gradient, thus driving spindle assembly [42]. RANBP1 in human somatic cells controls the spatial distribution and amount of mitotic RAN-GTP production, thereby ensuring the accurate execution of RAN-dependent mitotic events [34] (Figure 4).

## 4. Diseases Overview

During past decades, the role of RANBP1 in both nucleocytoplasmic transport and mitosis progression has emerged in literature, also supported by the advance of new biomolecular approaches, thus refining knowledges and completing the functional network of this small GTP-ases related protein. In the strictly bio-medical field, studies have been carried out directed at verifying whether RANBP1, either in gain or loss of function, may influence the transmission of diseases, and predispose to contract and alter the course of underlying diseases, in originally healthy patients. Certainties and novel insights related to neoplastic, central nervous system, viral and bacterial origin diseases and congenital malformations have been accumulating and therefore the main evidence in this regard will be explored in detail below (Figure 5).

### 4.1. RANBP1 and Neoplastic Diseases

The synthesis of nuclear-transport model converges on evidence that the assembly and disassembly of all complexes, consisting of cargo associated to a transport protein (e.g., karyopherin, importin, exportin, or transportin) are tightly controlled by the GTPase RAN, whose action is regulated by RCC1, RANGAP, and RANBP1. An interesting effect on cellular processes’ aging arise from the progressive extension of human life span and the undeniable causal relationship with the increasing incidence of neoplastic diseases in old age. mRNA levels for 6000 genes, derived from human dermal fibroblasts from young, mature, and elderly subjects, were analyzed. Approximately 1% of the mRNAs showed age-dependent changes: among the sixteen genes known to be involved in NCT (nucleo-cytoplasmic trafficking), as many as four showed age-dependent reductions, and, among them, the gene encoding for RANBP1. Therefore, at least in vitro cell lines, the reductions in RanBP1 in old age may lead to alterations in mitosis and in chromosome segregation. An age-dependent altered chromosomes segregation was also demonstrated in human lymphocytes. This, in vitro, evidence might suggest that the reductions in RANBP1 are a causal factor in the age-dependent reduction of the immune response and the increased aneuploidy. [43]. Of importance, it was established that the overexpression of RANBP1, induces the formation of multipolar spindles, thus demonstrating that these abnormal spindles originate from a loss of cohesion in mitotic centrosomes. Indeed, the excess of RANBP1 level induces the splitting of mother and son centrioles at the spindle poles. This aberrant cleavage requires microtubule integrity. Following the accrual of errors in this process, during mitosis, uneven segregation of chromosomes and production of aneuploid or polyploid cells occur, which might characterize both tumor initiation and progression [44]. Centrosomal duplication is assured not only by nucleocytoplasmic transport efficiency but also by RAN-CRM1-network mitotic recycling, since centrosomes are sensitive to inactivation of the nuclear export factor CRM1, which gives rise to supernumerary centrosomes and multipolar spindles. These mitotic abnormalities may also be a consequence either of, a mutation in RAN (which controls the interaction of transport receptors with their substrates) or RANBP1 dysregulation (which is the major RAN regulator and promotes the dissociation between RAN and CRM1). Therefore, the RAN-CRM1 complex negatively regulates the initiation of centrosomal duplication. Nucleophosmin (NPM), a ubiquitous phosphoprotein mainly localized in the nucleolus but able to shuttle between the nucleus and cytoplasm during the cell cycle (as it contains both NES and NLS sequence), may be a substrate of the RAN-CRM1 complex, defining a putative mechanism for centrosome duplication because of its ability to associate with unduplicated centrosomes, and to dissociate from them, following the initiation of centrosome duplication and DNA replication. The interaction between centrosomes and NPM is determined by the presence of a NES- and is sensitive to the inhibition of the CRM1-dependet nuclear export, induced by either leptomycin B (LMB) or *RANBP1* over-expression. LMB induces premature centrosome duplication in quiescent cells and this coincides with NPM dissociation from centrosomes. The NPM deficiency mediated by RANBP1 over-activity also results in the formation of supernumerary centrosomes and multipolar spindles, both conditions associated with most human cancer cells. This can be prevented by increased NPM on centrosomes promoted by the RAN-CRM1 complex [45]. Other evidence points to the following: elF4E (eukaryotic translation initiation factor) is a potent oncogene for the ability to alter the cytoplasmic side of PNC (nuclear pore complex) and reducing RANBP2, the major component of PNC cytoplasmic fibrils and inducing a concomitant and compensated RANBP1 increase, to achieve the maintenance of molecular trafficking that is no longer sufficiently ensured by RANBP2. This mechanism results in increased mRNA export and translation, and as matter a fact, oncogenic transformation. It is worth mentioning that ribavirin treatment leads to an elevation of RANBP2 and a reduction of RANBP1, leading to a reestablishment of the standard ratio, correlated to clinical remission in patients with AML. This evidence suggests that PNC reprogramming is a means by which oncogenes exploit the proliferative capacity of the cell [46]. The study on SINEs, a new generation of selective nuclear export inhibitors that act by CRM1 exportin blocking, is less toxic and more therapeutically manageable than other inhibitors, and shows the potent antileukemic activity of these drugs by inducing apoptosis in different molecular subtypes of the leukemic disease. KPT-251, a representative element of this new class of inhibitors, acts by binding CRM1 when it is stabilized in the trimeric CRM1-RAN-RANBP1 form. Once again, the latter’s involvement as a potential starter factor for severe proliferative hematologic diseases emerges [47]. In recent years, translational clinical works based on the analysis of *RANBP1* expression levels, and connections with in vivo tumor phenotypes and neoplastic advancement are largely growing. Recently, *RANBP1* expression was demonstrated to be highly increased in hepatocellular carcinoma (HCC) and related to advanced T stage and histopathological grade. For instance, it has been documented that high DNA methylation levels of *RANBP1* are significantly related to very poor overall survival (OS), thus highlighting how the *RANBP1* expression was associated with poor prognosis. An enrichment of the G2M checkpoint, Wingless and Int-1 (Wnt)-dependent cell signaling, and DNA repair pathways was documented in *RANBP1* high-expression phenotype by means of GSEA analysis, which also demonstrated that *RANBP1* was strongly over-expressed in the tumor-infiltrating lymphocyte compartment (tumor micro-environment), with particular regard to the (Th1) T helper cells component [48].

Remarkably in the human melanoma metastasis, four key components of the nucleocytoplasmic transport machinery—CRM1, RAN (RAN-GTPase), RANGAP1 and RANBP1, have been identified to be overexpressed. This pathway might be a potential therapeutic target in human melanoma, whose inhibition, by mediating a functional down-regulation of ERK1/2, results in a multitude of cellular context-dependent molecular events, thus triggering G1 arrest followed by massive apoptosis [49]. The network of small-GTPases RAN proteins, of which RANBP1 is one of the main effectors, has also been demonstrated involved in the onset and progression of epithelial ovarian cancer (EOC). The inference is obtained by finding that the high expression of RAN, and associated proteins XPO7 and TPX2, correlates with an increased tumor grading and is strongly related to the poor survival of patients. Likewise, in vivo, the decrease in RAN in a rat model, led to regression of the EOC tumor. RAN and its protein partners XPO7 and TPX2 may be used as biomarkers to stratify patients by both quality and life expectancy, their prognostic significance being attested. By an enriched patient cohort analysis for ovarian cancer, has been demonstrated that levels of cytoplasmic RAN were significantly higher in HG (high-grade tumors, poorly differentiated) tumors compared to LG (low-grade) tumors; concomitantly, the cytoplasmic staining intensity of RANBP1, Importin beta and nuclear localization of RCC1 were significantly increased in HG versus LG tumors. Such evidence would suggest that the network of small-GTPases-RAN proteins might be targeted to improve the survival of serous epithelial ovarian carcinoma [50].

In this context, which frames the role of RANBP1 in neoplastic deregulation and the ability of RANBP1 to mediate chemo- and radio-resistance phenomena in several tumor cell lines, primary cell models and in vivo plays a diriment role. This chapter develops from the evidence that the stress kinase SGK1 (serum- and glucocorticoid-regulated kinase 1) [51] implicated in neoplastic initiation and deregulation, through its ability to block several apoptotic pathways [52], mediate EMT processes and induce loss of cellular genomic control through MDM2-dependent p53 degradation [53], was a potent transcriptional and functional regulator of RANBP1. In the original evidence, it was demonstrated that SGK1, through a phosphorylation-dependent positive regulation of SP1, induces a potent and stable transcriptional activity of RANBP1 in colon carcinoma cell lines [54]. Already in the same p53 wild type colon carcinoma cell samples, it had been demonstrated that the over-expression of *SGK1* consistently corresponded to the over-expression of *RANBP1*, verified by two-dimensional and mass spectrometry approaches, and that conversely, the gene silencing of *SGK1* corresponded to the down-regulation of *RANBP1* expression. The discovery of the SGK1/SP1-dependent mechanism then clarified the regulatory molecular framework. This SGK1-dependent up-regulation of *RANBP1* corresponded to an advantage in escape from both mitotic death and early/late apoptosis, as well as to an instability of the mitotic spindle, configuring a full resistance to taxanes, of which the phenotype was reversed by the silencing of *SGK1* or blocking *SGK1* over-expression mediated by concomitant gene silencing of *RANBP1*. In this same context, *RANBP1* has also been demonstrated to be sensitive to SGK1-dependent regulation in several hepatocarcinoma cells and in xenograft mouse models for HCC (hepato-cellular carcinoma) [55]. In this system, it was also demonstrated how a benzo-pyrimidine-modified small molecule SI113 [56,57,58,59,60,61,62], an SGK1 inhibitor, through its SGK1 enzymatic blockade, is able to inhibit *RANBP1* expression. In brief, in the HCC context, the SGK1-RANBP1 regulatory axis has been shown to mediate the phenomena of chemo- and radio-resistance in cell lines as well as in vivo [55]. Interestingly, in HCC cell lines such as HUH7, the human embryonic kidney such as HEK293T and human prostate cancer cells such as PC3, primary human dermal fibroblasts such as HDFa, human ovarian carcinoma such as OVCAR3, and human glioblastoma cells such as ADF, a key role of the RAN/RANBP1/RANGAP1 system under SGK1 kinase regulation, in controlling the export of pre-miRNAs, was demonstrated [28]. The mechanism appeared to be generally verified, as it extended to entire libraries of miRNAs and extended to several normal and tumoral non-embryologically related cell models. Controlling the export of miRNAs determines their maturation and thus leads to the conclusion that the SGK1-RANBP1/RAN/RANGAP1 axis plays a key role in cellular epigenomic regulation [28].

Further insight into the topic of the RANBP1 role in colon cancer and microRNAs transport comes from evidence from the high expression of *RANBP1* in CRC (colon-rectal carcinoma) tissues of patients. The RANBP1 level was strongly associated with TNM stages and was an independent risk factor for poor prognosis. Functional in vitro and in vivo evidence have demonstrated that RANBP1 promotes CRC cell proliferation and invasion and inhibits CRC cell apoptosis. Low expression of *RANBP1* reduced the expression of the microRNAs (miRNAs) hsa-miR-18a, hsa-miR-183 and hsa-miR-106 by inhibiting the precursor miRNAs’ (pre-miRNAs) nucleoplasmic transport, thereby favoring their accumulation in the nucleus and thus reducing the maturation. It turned out that through miRNAs’ regulation, RANBP1 leads the expression of *YAP*, modulating the Hippo pathway. In turn, YAP plays as a transcriptional cofactor for the *RANBP1* transcription, in combination with the transcription factor TEAD4. Therefore, RANBP1 further promoted CRC progression by establishing a positive feedback loop with YAP. These results corroborate a RANBP1 biological role and mechanism in CRC, suggesting that it might be used as a potential diagnostic and therapeutic target in CRC [63]. In terms of further study on the topic, a mention should be made of the evidence for which 10% of colorectal cancer patients carry BRAF V600E and BRAF-related mutations, whose event is associated with a worse prognosis and poor outcomes towards systemic therapies. Pharmacological inhibition of NPM (nucleophosmin) effectively restored the susceptibility of vemurafenib-resistant BRAF-mutated CRC cells by down-regulating the expression and activity of c-Myc and consequently suppressing its transcriptional targets: RANBP1 and phosphoserine-phosphatase, that regulate centrosome duplication and serine biosynthesis, respectively [64]. Even CRM1 exhibits oncogenic features in colorectal cancer. One of its reversible inhibitors called S109 has been demonstrated to inhibit proliferation and arrest the cell cycle in colorectal cancers. Nuclear accumulation of RANBP1 can be used as a marker of CRM1 inhibition [65]. Moreover, by an immunofluorescence-based approach, it has been possible to monitor the subcellular localization of RANBP1 in ovarian cancer cells following the selective inhibitor S109 administration, thus strengthening the hypothesis of CRM1 as a possible target in cancer therapy, for its ability to block cell proliferation in ovarian cancer as well as colorectal cancer [66]. RANBP1 inactivation yields a hyperstable mitotic microtubule (MT) and promotes intra-mitotic apoptosis, mimicking the taxol effects on the MT-stabilizing. The influence of RANBP1 on spontaneous and taxol-induced apoptosis in mammary transformed cells has been studied. *RANBP1* downregulation by RNA interference activates apoptosis in several transformed cell lines regardless of their p53 status, but not in the caspase-3-defective MCF-7 breast cancer cell line. *RANBP1* silencing resulted in apoptotic boost, and this response was caspase-3 dependent. These results indicate that RANBP1 may modulate the outcome of therapeutic protocols by affecting the MTs’ counterpart; this may play a key role in the targeted treatment of breast cancer, where taxanes play the role of first-line chemotherapeutics [67]. Despite the great clinical usefulness of paclitaxel (taxol), there is a high variability in the clinical response to the drug, limiting the use. Unfortunately, the mechanism behind this variability remains unknown. An engrafted tumor model was created, in order to in vivo evaluate the tumor cells’ sensitivity to paclitaxel. In vitro and in vivo assays demonstrated that *CD147* silencing sensitized tumor cells to paclitaxel treatment, whereas *CD147* over-expression protected tumor cells from caspase-3-mediated apoptosis induced by paclitaxel, regardless of p53 status. The intracellular domain of CD147 (CD147ICD) is essential for CD147-regulated paclitaxel sensitivity. RANBP1 mediates CD147-regulated microtubule stability and dynamics and in turn, the paclitaxel efficacy. In brief, CD147 regulates the response to paclitaxel by interacting with the C-terminal tail of RANBP1, and targeting CD147 may be another promising strategy to prevent paclitaxel resistance [68]. Recently, evidence has been accumulating to support that the overexpression of RAN GTPase protein-network contributes to chromosomal instability (CIN) and worsens the prognosis of breast cancer patients. Contrary to what one would expect, only marginal correlation has been found with overexpression of components strictly involved in spindle assembly and nuclear import. Rather clear, however, is the link between the overexpression of nuclear export components, such as RANBP1 and the poor prognosis of breast cancer. These components could be investigated as potential new targets for therapy [69] (Table 1).

### 4.2. RANBP1 and Central Nervous System Diseases

The RAN-dependent nucleocytoplasmic transport machinery is essential and step-limiting for the establishment of cortical neuron polarity differentiation. RANBP1 regulates axon specification and dendritic arborization of cultured neurons in vitro and radial neural migration in vivo. During axon-genesis, RANBP1 controls the cytoplasmic levels of LKB1/PAR4, involved in neuronal polarity, whose activity is dependent on the nuclear export machinery [70]. Moreover, RANBP1 is required to establish anterior–posterior polarity so that neural crest (NC) cells are able to chemotax. In addition, at the regard of chemotactic activity, RANBP1 require the activation of the kinase LKB1/PAR4. The RAN/RANBP1-dependent LKB1 nuclear export regulation is a required step to establish anterior–posterior polarity, and thus chemotaxis, during the collective migration of NCs [71].

Originally, RANBP1 was found in cDNA samples from frontonasal masses, branchial arches, limbs, and heart and wherever ridge-derived mesenchymal cells are concentrated (inductive signaling sites). In addition, *RANBP1* is demonstrated to be expressed in cells of tissues within or adjacent to the cerebellum, and along the caudal neural tube [72]. In humans, *RANBP1* is located at the chromosome 22q11.1, which is commonly deleted in the velo-cardio-facial syndrome, also called Di George syndrome. This syndrome arises as de novo in 90% of cases. In other circumstances, the genetic defect can be transmitted by an affected parent (dominant autosomal). When transmitted, it affects one in every 4000 live births. Characteristic symptoms include heart defects, facial abnormalities, cleft palate, hypocalcemia (often auto-regressive), liver abnormalities, immunodeficiency T, learning disabilities (framed within the autism spectrum) and increased risk of mental illness. Recently, it has been made possible for one to identify more than 200 different phenotypic features. The etiopathogenetic sense of canonical phenotypic variety might depend on the RANBP1 involvement in the formation of the cilia, a structure that performs essential functions during embryogenesis [73]. More recently, the possible association between 22q11.2 deletion and an early-onset, L-dopa-responsive form of Parkinson’s disease, has also been described. The finding of this genetic abnormality could be used as a potential predictive marker [74]. At this regard, it has to be considered that RAN GTPase is involved in the control of neuronal growth in mammals and retrograde signaling of nerve axons after injury [75]. Several *RANBP1* polymorphisms (RanBP1-ht2) have been described in subjects with SPEM (abnormality of smooth pursuit eye movement), a major neurophysiological alteration in patients with schizophrenic disorder. However, no *RANBP1*-haplotype has so far appeared to be directly associated with the risk of schizophrenia [76]. Although RANBP1 has still not a specific known function during brain development, it is highly expressed in the ventricular/subventricular zone of the developing forebrain. This would explain its role in microcephaly and cortical connectivity disorders, such as autism and attention deficit syndrome, because of its selective power to disrupt the cortical projection neuron [77]. Abnormality in mGluR5 (metabotropic glutamate receptor 5)-mediated signaling is involved in the autism spectrum disorders (ASD), tuberous sclerosis and fragile X syndrome pathophysiology. The simultaneous finding of autism spectrum disorders, in individuals haploinsufficient for the mGluR-associated RANBP1 gene, could provide a genetic relationship between apparently unrelated genetic and environmental forms of ASD [78].

### 4.3. RANBP1 in Viral and Bacterial Lifecycle

The Rev-HIV holds an RNA-binding domain and a nuclear export signal (NES). Mutations in the NES domain are found to be trans-dominant (TD) and result in the inhibition of Rev’s nucleocytoplasmic export, arising in HIV inhibition. Functional and structural analogies have been demonstrated between the RANBP1-NES-domain and NES of HIV-1 Rev. Experimental evidence leads one to hypnotize the ability of NES-RANBP1 to replace the NES of HIV-1 Rev. In brief, RANBP1-NES signals and Rev functionally cross-interfere, showing a competition for a common export pathway [79]. A RANBP1 role in the HIV1 Rev protein export can be inferred through evidence conducted on HERC5. HERC5 is a ubiquitin-protein ligase or E3, encoded in humans by *HERC5* gene. This protein plays as E3 ligase for ISG15, a protein induced by interferon type 1 (IFN) that inhibits the replication of several viruses, including HIV-1, influenza A virus and papilloma virus. More specifically, HERC5 inhibits HIV-1 particle production by targeting the nuclear export of Rev-dependent RNA. HERC5, through its RCC1-like amino-terminal domain, inhibits the production of HIV-1 particles by affecting the nuclear export of Rev-dependent RNA. This mechanism is related to the effects of the interaction between HERC5 and RAN, causing a reduction in intracellular levels of RANGTP and the latter’s ability to interact with RANBP1 [80]. RANBP1 can displace Nup (nucleoporins) and form a ternary RANBP1/RANGTP/CRM1 complex that can be disassembled by RANGAP via GTP hydrolysis. RANBP1/RANGTP/CRM1 can also be disassembled, without affecting GTP hydrolysis, by the nucleotide exchange factor in the RAN-GEF activity. Recycling of a RAN/RANGEF complex by GTP and Mg2+ is stimulated by both CRM1 and Rev, allowing the reformation of a Rev/CRM1/RANGTP complex [81]. RANBP1 also plays an important role in the replication and metabolism of some specific bacteria. Studies on *Legionella pneumophila*, the causative agent of Legionnaires’ disease, have provided a better understanding of the RANBP1 role. *Legionella pneumophila* uses the Icm/Dot type IV secretion system (T4SS) to form in phagocytes a distinct “Legionella-containing vacuole” (LCV), which intercepts endosomal and secretory vesicle trafficking. RAN and RANBP1 localize to LCVs and promote intracellular growth of *L. pneumophila*. Moreover, the *L. pneumophila* protein LegG1, containing a putative RCC1 RAN guanine nucleotide exchange factor (GEF) domains, in an Icm/Dot-dependent manner, accumulates on LCVs. In *L. pneumophila*-infected amoebae or macrophages, RAN and RANBP1 localize to LCVs, and the GTPase is activated by the Icm/Dot substrate LegG1. RAN activation by LegG1 leads to microtubule stabilization and promotes intracellular pathogen vacuole motility and bacterial growth, as well as chemotaxis and migration of *Legionella*-infected cells [82,83].

### 4.4. RANBP1 in Teratogenesis and Infertility

Experiments conducted on the zebrafish embryo have identified a RANBP1 cDNA, encoding for a polypeptide, strongly resembling to the *Xenopus* and human form. In zebrafish, *RanBP1* mRNA has maternal transcript lineage and is ubiquitously expressed at all embryonic stages. Overexpression of the protein showed no effect on embryogenesis. In contrast, the down-regulation of *RanBP1*, increased the mortality rate and frequency of defects, achieved by double-stranded RNA inoculation (RNA interference), although the low specificity of the latter did not fully confirm the deregulatory interference [84]. Rather important, however, in the field of embryonic development and teratogenesis is the demonstration that Knock-out mice for *RanBP1* manifest unexpected viability correlated with male infertility due to the blockade of spermatogenesis, likely managed by concomitantly *RanBP2* down-regulation during spermatogenesis. Conversely, the knock-down of *RanBP2* caused cell death only in MEFs derived from *RanBP1* knock-out mice and not in control mice. This indicated that the simultaneous depletion of *RanBP1* and *RanBP2* definitely impairs germ cell viability [85]. At this aim, RanBP1 may be considered as new biomarker, enabling a reliable and noninvasive diagnosis of azoospermia. Different types of azoospermia are described: obstructive (OA) and nonobstructive (NOA) azoospermia, hypo-spermatogenesis subtypes (Hyp), and Sertoli cell-only syndrome (SCO). Using a comparative proteomics approach, starting from testicular tissue, several pathways associated with azoospermia and a number of testis- and germ cell-specific proteins have been identified, showing the potential to recognize the spermatogenesis failure subtype. Among these proteins, RANBP1, H1-6 and TKTL2 showed superior clinical power for quantitative discrimination between OA, Hyp and SCO [86].

## 5. Conclusions

RANBP1 has been demonstrated to be essential and step-limiting in several molecular pathways and clear pathophysiological alterations. The continuous emergence of biochemical and clinical evidence that fixes in RANBP1, the junction of key physiological and pathological pathways in the cellular life cycle, makes clear the need to develop molecular modulators of this protein with possible clinical uses. Intriguingly, not only inhibitors but also positive modulators might find a place in mediating complex phenotypes that are altered by either up- or down-regulation of *RANBP1*.

## Figures and Tables

**Figure 1 cancers-15-00486-f001:**
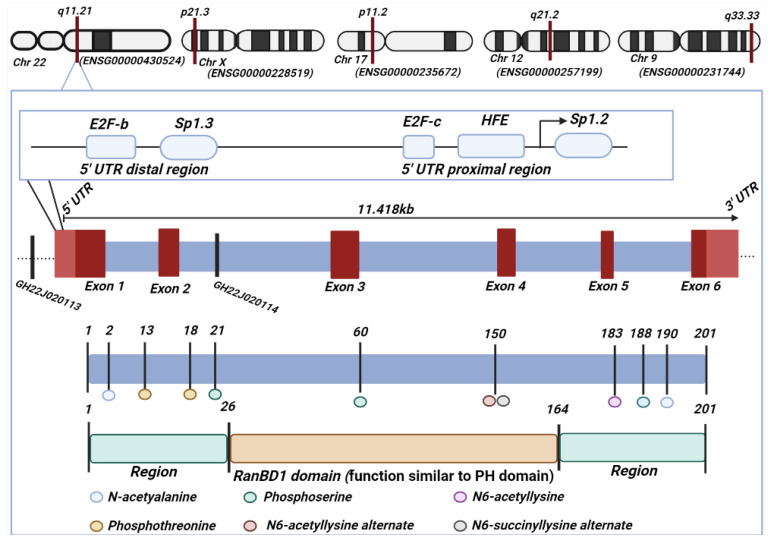
Structure of RANBP1. Gene’s chromosomal localization, related transcript, protein/domain and post-translational modifications. Schematic representation.

**Figure 2 cancers-15-00486-f002:**
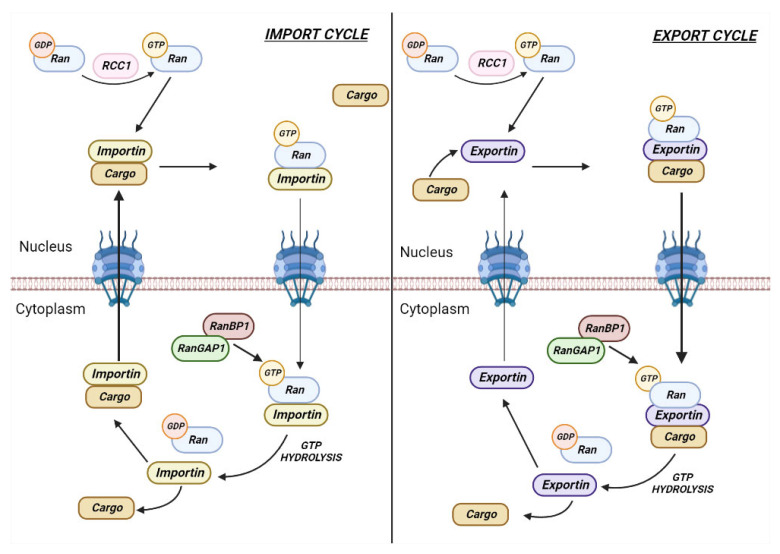
RANBP1 mediates nuclear transport through the RAN complex (arbitrary putative model). Nucleus import and export cycles and model for Importins/Exportins (carrier proteins) recycling. Importins (**left**) bind to cargo in the cytoplasmic side and allow interactions with the nuclear pore complex to promote translocation of the import complex into the nucleus. RanGTP in the nucleus binds to importin and favors cargo release. The importin-RanGTP complex is then recycled into the cytoplasm, where RanGTP is detached from importin through RanBP1-RanGAP1 dependent GTP hydrolysis. The export cycle is comparable (**right**) but RanGTP induces cargo binding in the nucleus. After removal of RanGTP from the complex through RANBP1/RANGAP1-dependent hydrolysis of GTP in the cytoplasm, exportins separates from cargo and the empty receptor is recycled into the nucleus. Freely inspired and adapted from [26].

**Figure 3 cancers-15-00486-f003:**
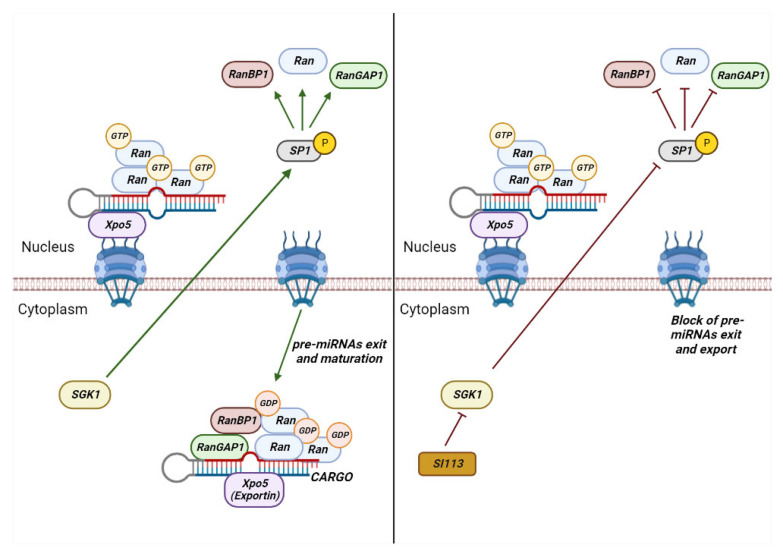
SGK1 regulates the RAN/RANGAP1/RANBP1 axis, influencing pre-miRNAs nuclear export and maturation. The figure depicts the SGK1-dependent pre-miRNAs nuclear export regulation, by modulating the RAN complex in SP1-dependent manner. Conversely, the SGK1 activity inhibition affects pre-miRNAs’ export and maturation in the cell cytoplasm (see also Section 4.1).

**Figure 4 cancers-15-00486-f004:**
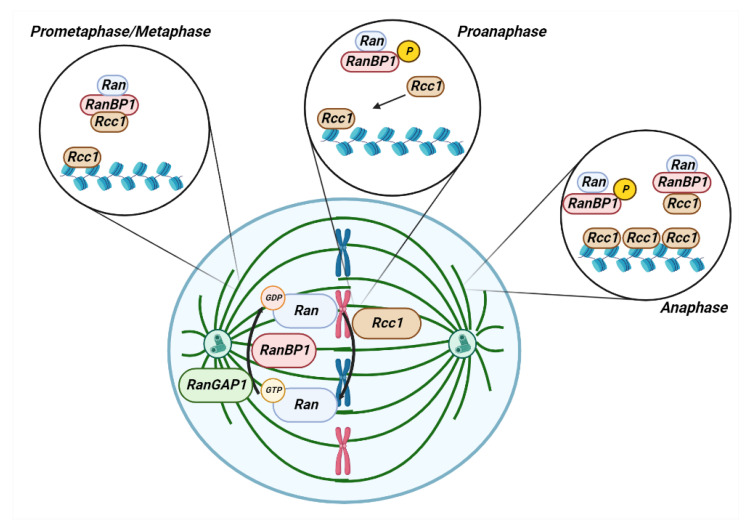
The RCC1/RAN/RANGAP1/RANBP1-axis in mitotic stability. Prometaphase: RCC1 fluctuates between an active state (chromatin bound pool) and an inactive state, bound to RANBP1 and nucleotide-free-RAN (RRR complex). Anaphase Onset: upon Ser 60 RANBP1 phosphorylation, RCC1 detachment from the RRR-complex is realized. The free RCC1 is then engaged on chromatin. Anaphase: Increased chromatin-bound RCC1 levels promote switch from Ran-GDP to Ran-GTP on chromatin. Interphase (not shown): nuclear envelope re-formation physically splits RCC1 from RanBP1, thus preventing RRR complex assembly and RCC1 inhibition. Freely based from [42].

**Figure 5 cancers-15-00486-f005:**
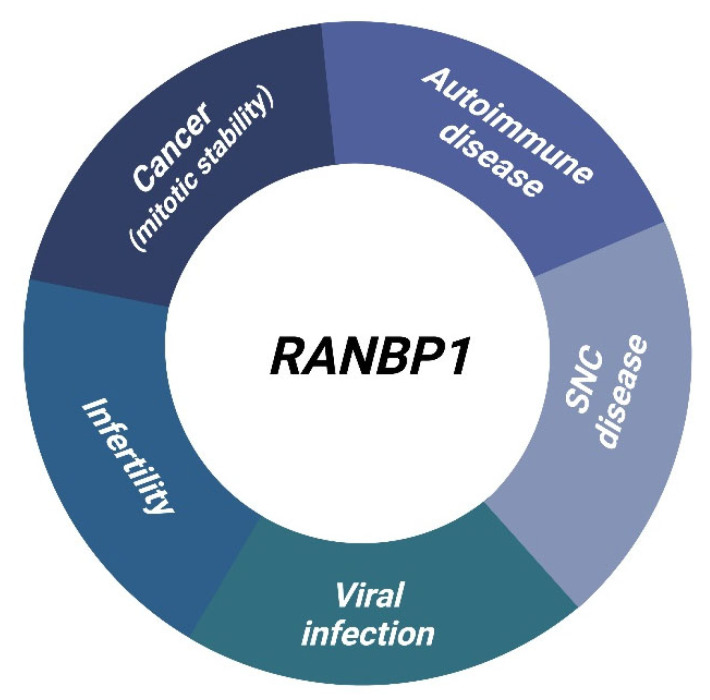
RANBP1 involvement in human diseases.

**Table 1 cancers-15-00486-t001:** RANBP1 involvement in human cancers.

Cancer Type	References
Ovarian Cancer	[28,50,61,65,66,67,68]
Prostate Cancer	[28]
Glioblastoma	[28]
Osteosarcoma	[46,67]
Acute Monocytic Leukemia	[47]
Hepatocellular Carcinoma	[28,48,55]
Melanoma	[49]
Colon Cancer	[54,63,64]
Brest Cancer	[67,69]
Lung Cancer	[68]

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
