# Peer review of "RANBP1 (RAN Binding Protein 1): The Missing Genetic Piece in Cancer Pathophysiology and Other Complex Diseases"

_cancers, 2023, doi:10.3390/cancers15020486_

Round 1

Reviewer 1 Report

The manuscript “RANBP1 (RAN binding protein 1): the missing genetic piece in cancer pathophysiology and other complex diseases” is a review delving into the multifaceted roles of the RANBP1 factor. Being RANBP1 deeply involved in nucleo-cytoplasmic trafficking, its function is involved in several patho-physiological conditions, all well described in the text.

The review is exhaustive and quite explicative, ranging from the physiological role of this factor to the pathological states that involve its dysfunction. On these bases, it will be quite useful for Cancers readers and thus worth of publication.

Nevertheless, this reviewer would like to raise some queries to be addressed.

·       Can the authors precise if there is any relationship between RANBP1 and inflammation, especially in the tumor microenvironment?

·       There are minor typo errors to correct, e.g., line 98.

·       Figure 3 introduces SGK1 and SI113 that are discussed far below

·       The numbering of the figures does not respect the sequence of their appearance in the text.

·       Lines from 401 to 433 are a duplicate of the previous paragraph.

·       DOI in Reference 5 is probably incorrect.

Author Response

First Reviwer:

  • Can the authors precise if there is any relationship between RANBP1 and inflammation, especially in the tumor microenvironment?
  1. thank you for this very interesting observation. There is currently no work that totally elucidates the role of RANBP1 in the inflammatory and tumor microenvironment context, however recently a paper by Wei et al in J. Gastrointest Oncol from 2021 (citation #49 in our review) approaches the topic for the first time. The data in our opinion are inconclusive and in some ways controversial, however given the descriptive and non-systematic nature of this review we have now accounted for this more adequately.
  • There are minor typo errors to correct, e.g., line 98.
  1. thank you very much. The typo has now been corrected
  • Figure 3 introduces SGK1 and SI113 that are discussed far below
  1. thank you very much for this comment. the figure we believe should remain there for the logic of the narrative flow, however as per your suggestion we have now supplemented the text by anticipating in a summary way in the nuclear transport section what we then address more fully in the next paragraph.
  • The numbering of the figures does not respect the sequence of their appearance in the text.
  1. Thank you very much, the point has now been addressed
  • Lines from 401 to 433 are a duplicate of the previous paragraph.
  1. Infinite thanks for this relief. During the process of formatting and transferring text from word file to cancers file, an erroneous duplication of text occurred, now fixed.

Reviewer 2 Report

It is a review that tries to shed light on the role of RANBP1 in the caner pathophysiology  and other complex diseases. Audia et al have done exhaustive work on this protein, very well presented and written. I am convinced that this is very interesting work both for specialists in the field and for Cancers readers in general. However, the volume of information is very high and makes it difficult to follow up and understand the review.  I think that sections 4.2, 4.3 and 4.4, even though they are interesting, should be eliminated. So the review focuses on cancer. This will enrich the manuscript and would be appreciated by the reader.

Author Response

Second Reviwer

It is a review that tries to shed light on the role of RANBP1 in the caner pathophysiology  and other complex diseases. Audia et al have done exhaustive work on this protein, very well presented and written. I am convinced that this is very interesting work both for specialists in the field and for Cancers readers in general. However, the volume of information is very high and makes it difficult to follow up and understand the review.  I think that sections 4.2, 4.3 and 4.4, even though they are interesting, should be eliminated. So the review focuses on cancer. This will enrich the manuscript and would be appreciated by the reader.

  1. Thank you very much for this judgment. I agree with you in principle, however, the authors' willingness to leave the presentation of the text this way stems from several points:

(a) it is the first review in 30 years that the gene has been cloned, which explains the impressive amount of data collected but also the importance of having a reference review for all researchers approaching the topic

  1. b) since the title envisages the role of the protein in cancer but also in diseases of the complex traits, it is necessary, also in a reduced way as we have done, to provide hints of understanding of the role of RANBP1 in other pathological contexts.

(c) To leave out two or three small topics, since there are still not enough publications for a dedicated review, would be to erase these authors from the albeit important history of the gene

(d) Molecoar and neoplastic rather than viral aspects far from being separate fields often have points of union and shared developments.

Reviewer 3 Report

The review by Audia and colleagues focuses on the role of RANBP1 and its gene product (RAN binding protein 1) in the RAN (a small GTPase member of the RAS superfamily) signaling network.  The manuscript has many weaknesses both in form as well as in content. These weaknesses make the review unacceptable. A list of some of the detected weaknesses is below to help the authors to fix them for future submissions.

Content deficiencies.

The review as is written is too narrow and only the researchers working on topics directly related to this gene will be able to grasp the details described.  For instance, the review should describe first in the introduction RAN and its signaling network.  They only briefly address this well within the review. Also, the review should mention in the introduction that these proteins belong to the RAS superfamily of small GTPases. Then, the reader would be in better position to follow the details of the paper.

Another conceptual problem is the over statements made supporting the claimed important role of the gene and its encoded protein in “almost every aspect of mammalian cell biology” (Simple Summary). Also in the Abstract, the conclusion that “not only, RANBP1 appears to be a causative point of several pathological conditions but also a possible future target of innovative therapies” is no more than a gross speculation without experimental support. “Causative points” of pathological conditions have not been demonstrated for RAN or its related proteins. Just a point to be considered and discussed would be the apparent absence of somatic mutations in the gene in cancer, or germ line mutations in other pathologies, in contrast with the RAS (onco)genes. And RAS gene mutations are not causative, but only contributing in cancer.

Form deficiencies.

The figures are non informative and confusing.

Figure 1 depicts the localization in chromosome 22 of RANBP1 and protein domains and posttranslational modifications. In the text it is written “Sixty-three different regulatory elements for RANBP1, such as promoters and enhancers, have been identified along its sequence, among which GH22J020114 and GH22J020113 are more frequently associated with its sequence [2]. The protein structure contains a pleckstrin homology (PH) domain involved in protein-protein interaction with RAN and in cell signaling [3]. RANBP1 counts multiple pseudogenes encoded on chromosomes 17, 9, 12 and X, showing a wide variety of alternative splicing, resulting in multiple transcript variants [4]. All these genetic and structural information are depicted in Figure 1.”

However, there is no depiction of the different enhancers and promoters, the PH domain is not shown, and the figure shows instead a RanBD1 domain, that suggest the protein interacts with the gene and not with the protein.  Also, the two Region, at 5’ and 3’ are not explained. The pseudogenes and their transcripts are not part of the figure or their relevance discussed in the review.

Figure 2 illustrates the role of RANBP in nucleus-cytoplasm traffic, but there are many unexplained mechanisms, and fails to show the claimed “central role of RANBP1”. RANBP1 is only one box (in each side of the figure) while other factors are present in several boxes in and out of the nucleus. The figure shows left and right pathways, but they are nearly identical, with only “exportin” or “importin” being differentially present or absent. There is no description of what makes them to be present or absent.  And according to the figure, there is no difference between export and import, to or from the nucleus.

In the text (lines 116-117) “The RANBP1 release from nucleus is conditioned by a NES, …”, but there is no description of what NES is.    

Figure 3 focuses on the export from the nucleus of microRNAs, but again the schematic representations are confusing, and the figure legend is insufficient as well as the description in the Results section. There is no description of what “cargo” is. Also, in the text is used miRNA but in the figure Mirna.  And there are grammatical errors in the text: line 152, “releases” when should be “release”.  There is a mention of “exporting” in regards to Figure 3, but there is no “exporting” in Figure 3.

Figure 4 also is also insufficiently explained in its legend and confusing in the corresponding Results section that is full of grammatical errors.

Figure 5 is overstretching and simplistic.

Another unacceptable deficiency is in regards to the list of references.  Refs 20, 21, 24, 25, 28, 31, 37, 39, 44 and 80 are incomplete, without Journal information. And ref, 61 has an aberrant insertion: “<Sup>64</Sup>CuCl< Sub>2</Sub>” These deficiencies are diagnostic of a rushed and careless writing that are unacceptable.    

Other typos and grammatical errors:

“RANBP1 is RAN major effector, cause of the ability to interact only with RANGTP, but not with RANGDP, ...” (lines 94-95)

“…holding the latter complex in a less responsive sate for guanine nucleotide exchange on..” (line 101).

“… whereas leading to over-expresses this in G1/S phase” (line 180) in unclear.

“… triphosphate gaunidic nucleotides of RAN.” Should be guanidic.

“… phase, reach a peak in metaphase and then sharply decrease…” (line 199). Reaches and decreases.

“Interestingly, RANBP1 overexpression be held responsible for abnormalities in mitosis and defects in spindle assembly [37].” Incorrect grammar.

Author Response

Referee 3

 We thank this referee for his timely, detailed and interesting remarks and also for such a kind and polite way of expressing them. A point-by-point rebuttal follows in detail:

Content deficiencies.

The review as is written is too narrow and only the researchers working on topics directly related to this gene will be able to grasp the details described.  For instance, the review should describe first in the introduction RAN and its signaling network.  They only briefly address this well within the review. Also, the review should mention in the introduction that these proteins belong to the RAS superfamily of small GTPases. Then, the reader would be in better position to follow the details of the paper.

  1. I thank the referee for this thoughtful and timely observation. Since this is a complex point I would   divide my argument into several sub-responses.

1) when I have been invited for this review, being quite experienced in the topic, I intended to propose a deliberately descriptive review because of the following arguments, which in my opinion cannot be evaded:

(a) it is the first review in 30 years on RANBP1, so rather than critically argue the evolution of the literature about this gene, it was important to report, synthesize, and harmonize it into a single narrative.

(b) many of the papers on RANBP1, are the work of the correspondent author's group or other groups with which the correspondent collaborates. So critical arguments about the work could constitute an unacceptable conflict of interest, vice versa a form of descriptive review as well as informing and synthesizing all the literature on the topic avoids an unbearable and presumptuous conflict of interest.

(c) the editorial project, presented in this form to editorial office of Cancers in advance, has been evaluated interesting

2) The present review is rather long and difficultly single parts of this can be removed, that is why the part of RAN and its protein-network rather than presented in a single paragraph was extensively discussed in the various sub-sections of interest.  In accordance with the remarks of this referee, we have now added further statements, both to clarify membership in the RAN network and to specify membership in the RAS super-family, this both in summery and abstract and in introduction.

Another conceptual problem is the over statements made supporting the claimed important role of the gene and its encoded protein in “almost every aspect of mammalian cell biology” (Simple Summary). Also, in the Abstract, the conclusion that “not only, RANBP1 appears to be a causative point of several pathological conditions but also a possible future target of innovative therapies” is no more than a gross speculation without experimental support. “Causative points” of pathological conditions have not been demonstrated for RAN or its related proteins. Just a point to be considered and discussed would be the apparent absence of somatic mutations in the gene in cancer, or germ line mutations in other pathologies, in contrast with the RAS (onco)genes. And RAS gene mutations are not causative, but only contributing in cancer.

On this point I fully agree with the referee and apologize for the misunderstanding. the point has been removed and the statement scaled down in scope and expectation in accordance with the referee's relief

Form deficiencies.

 The figures are non-informative and confusing.

Many of the presented figures were inspired and modified from high-impact and seminal papers. This fact has now been better specified. However, the changes requested by the referee have been accepted and argued in detail below

Figure 1 depicts the localization in chromosome 22 of RANBP1 and protein domains and posttranslational modifications. In the text it is written “Sixty-three different regulatory elements for RANBP1, such as promoters and enhancers, have been identified along its sequence, among which GH22J020114 and GH22J020113 are more frequently associated with its sequence [2]. The protein structure contains a pleckstrin homology (PH) domain involved in protein-protein interaction with RAN and in cell signaling [3]. RANBP1 counts multiple pseudogenes encoded on chromosomes 17, 9, 12 and X, showing a wide variety of alternative splicing, resulting in multiple transcript variants [4]. All these genetic and structural information are depicted in Figure 1.” However, there is no depiction of the different enhancers and promoters, the PH domain is not shown,  and the figure shows instead a RanBD1 domain, that suggest the protein interacts with the gene and not with the protein.  Also, the two Region, at 5’ and 3’ are not explained. The pseudogenes and their transcripts are not part of the figure or their relevance discussed in the review.

thank you very much for your observation. The figure has been modified in accordance with this referee's timely remarks and now represents the points discussed. Regarding the PH domain, it was perhaps mistakenly taken for granted by the writer, that in RANBP1 the PH domain, at least in its functions rather than structure was encompassed and vicariated by the BD1 domain, which in addition to binding tri/diphosphate nucleotides acts as a protein-protein interactor. This fact is now better described in the figure and in the text.

“RanBP1 interacts specifically with GTP-charged Ran. RanBP1 does not activate GTPase activity of Ran, but does markedly increase GTP hydrolysis by the RanGTPase-activating protein (RanGAP1). In both mammalian cells and in yeast, RanBP1 acts as a negative regulator of Regulator of chromosome condensation 1 (RCC1) by inhibiting RCC1-stimulated guanine nucleotide release from Ran. In addition to Ran, RanBP1 has been shown to interact with Exportin-1 and Importin subunit beta-1 which docks the NPC at the cytoplasmic side of the nuclear pore complex. RanBP1 contains a single RanBD. The RanBD is present in RanBD1, RanBD2, RanBD3, Nuc2, and Nuc50. Most of these proteins have a single RanBD, with the exception of RanBD2 which has 4 RanBDs. Ran is a Ras-like nuclear small GTPase, which regulates receptor-mediated transport between the nucleus and the cytoplasm. RanGTP hydrolysis is stimulated by RanGAP together with the Ran-binding domain containing acessory proteins RanBP1 and RanBP2. These accessory proteins stabilize the active GTP-bound form of Ran. The Ran-binding domain is found in multiple copies in Nuclear pore complex proteins. RanBD shares structural similarity to the PH domain, but lacks detectable sequence similarity”.

cd13179: RanBD_RanBP1

 Figure 2 illustrates the role of RANBP in nucleus-cytoplasm traffic, but there are many unexplained mechanisms, and fails to show the claimed “central role of RANBP1”. RANBP1 is only one box (in each side of the figure) while other factors are present in several boxes in and out of the nucleus. The figure shows left and right pathways, but they are nearly identical, with only “exportin” or “importin” being differentially present or absent. There is no description of what makes them to be present or absent.  And according to the figure, there is no difference between export and import, to or from the nucleus.

I apologize, you are absolutely right. The figure has been revised but more importantly, the caption has been extensively changed, explaining in detail the import/export mechanism, importins/exportins recycling system  and the role of RANBP1 in this. The difference between the left and right side of the figure has also been clarified.

In the text (lines 116-117) “The RANBP1 release from nucleus is conditioned by a NES, …”, but there is no description of what NES is.    

thank you very much it has now been better clarified that we are talking about generic NES. That is, ranbp1 acts on the NES domains of cargo proteins in the transport complex. 

Figure 3 focuses on the export from the nucleus of microRNAs, but again the schematic representations are confusing, and the figure legend is insufficient as well as the description in the Results section. There is no description of what “cargo” is. Also, in the text is used miRNA but in the figure Mirna.  And there are grammatical errors in the text: line 152, “releases” when should be “release”.  There is a mention of “exporting” in regards to Figure 3, but there is no “exporting” in Figure 3.

Good point, Thank you. the figure has now been modified to better convey the mechanism of pre-miRNAs nuclear exporting. Also it has been clarified that in this system, the pre-miRNAs represent the cargo and that Exportin-5  is the carrier. On the scientific value of the figure let me point out that I am well acquainted with the subject since the figure has been remodeled and repurposed from my own paper, "SGK1 affects RAN/RANBP1/RANGAP1 via SP1 to play a critical role in pre-miRNA nuclear export: a new route of epigenomic regulation" SCIENTIFIC Reports | 7:45361 | DOI: 10.1038/srep45361

Figure 4 also is also insufficiently explained in its legend and confusing in the corresponding Results section that is full of grammatical errors.

I agree with you and thank you. The figure Legend has been completely changed and the corresponding paragraph simplified and rewritten.

Figure 5 is overstretching and simplistic.

In this case I feel I am keeping to the point. this figure is introductory to the pathology chapter and together with the next table serve only as a general orientation for the reader and almost as an index and thus is deliberately simplistic. Figures of this type and with this graphic form are routinely found in many reviews and with the same function. However, I thank the referee again for the polite suggestion.

Another unacceptable deficiency is in regards to the list of references.  Refs 20, 21, 24, 25, 28, 31, 37, 39, 44 and 80 are incomplete, without Journal information. And ref, 61 has an aberrant insertion: “<Sup>64</Sup>CuCl< Sub>2</Sub>” These deficiencies are diagnostic of a rushed and careless writing that are unacceptable.    

             Regarding the last sentence, I will refrain from commenting and only respond that corrections have been made.

     Other typos and grammatical errors:

            thank you very much, now, these have been corrected

Round 2

Reviewer 2 Report

I still think that it would be better if the review were reduced, considering that this magazine deals with cancer related topics. In my view the editors should make the decision.

Author Response

Thank you very much. I greatly respect your point of view, and holding my position does not pertain to the unreasonableness of your argument as much as it does to the editorial design of the originally submitted review and the title itself. As you suggest we rely on the editor to make the final decision.

Reviewer 3 Report

See attached responses to the authors' rebuttal.

Author Response

Please, enclosed find my answers to this referee.

Round 3

Reviewer 2 Report

The review has improved substantially, so I suggest be accepted in present form for publication in Cancers.

Author Response

thank you very much